# Recent Advancements and Future Prospects of Noble Metal-Based Heterogeneous Nanocatalysts for Oxygen Reduction and Hydrogen Evolution Reactions

**Dinesh Bhalothia [1], Lucky Krishnia [2], Shou-Shiun Yang [1], Che Yan [1] , Wei-Hao Hsiung [1], Kuan-Wen Wang [3] and Tsan-Yao Chen [1,4,5,6,*]**

[1] Department of Engineering and System Science, National Tsing Hua University, Hsinchu 30013, Taiwan; dinesh@mx.nthu.edu.tw (D.B.); kent108011544@gapp.nthu.edu.tw (S.-S.Y.); s106011506@gapp.nthu.edu.tw (C.Y.); teddy01070107@gapp.nthu.edu.tw (W.-H.H.)
[2] Amity Centre of Nanotechnology, Amity University Haryana, Panchgaon, Gurugram-122413, India; lkrishnia@ggn.amity.edu
[3] Institute of Materials Science and Engineering, National Central University, Taoyuan City 32001, Taiwan; kwwang@ncu.edu.tw
[4] Institute of Nuclear Engineering and Science, National Tsing Hua University, Hsinchu 30013, Taiwan
[5] Hierarchical Green-Energy Materials (Hi-GEM) Research Centre, National Cheng Kung University, Tainan 70101, Taiwan
[6] Department of Materials Science and Engineering, National Taiwan University of Science and Technology, Taipei 10617, Taiwan
\* Correspondence: chencaeser@gmail.com; Tel.: +886-3-5715131 or +885-3-5720724

**Abstract:** The oxygen reduction reaction (ORR) and hydrogen evolution reaction (HER) both are key electrochemical reactions for enabling next generation alternative-power supply technologies. Despite great merits, both of these reactions require robust electrocatalysts for lowering the overpotential and promoting their practical applications in energy conversion and storage devices. Although, noble metal-based catalysts (especially Pt-based catalysts) are at the forefront in boosting the ORR and HER kinetics, high cost, limited availability, and poor stability in harsh redox conditions make them unfit for scalable use. To this end, various strategies including downsizing the catalyst size, reducing the noble metal, and increasing metal utilization have been adopted to appropriately balance the performance and economic issues. This mini-review presents an overview of the current state of the technological advancements in noble metal-based heterogeneous nanocatalysts (NCs) for both ORR and HER applications. More specifically, we focused on establishing the structure–performance correlation.

**Keywords:** hydrogen evolution reaction; nanocatalysts; electrochemical catalysis

## 1. Introduction

The continuously increasing imbalance between energy supply and demand along with adverse climatic issues have reinforced the importance of implementing a sustainable and environmentally benign energy economy. Recently, the International Energy Agency reported that the global energy demand will increase by nearly 30% by the end of 2040. Meanwhile, the $CO_2$ emissions will hit 35.7 Gt year$^{-1}$ by 2040 [1]. In this context, fuel cells are believed to be an ultimate solution for the future energy supply owing to their high energy conversion efficiency with zero emissions [2]. However, the sluggish kinetics of the oxygen reduction reaction (ORR) on the cathode side is a major hurdle for their practical and scalable usage; hence, we need highly efficient electrocatalysts to boost ORR kinetics [3]. At the same time, hydrogen ($H_2$) has also emerged as an ultimate ideal energy carrier

by virtue of it having the highest gravimetric energy density (142 MJ kg$^{-1}$) of all the carbonaceous species and zero emissions [4]. On Earth, hydrogen is not found in free form and is mainly available in compounds (e.g., hydrocarbons and water), hence, the effective production of hydrogen is of paramount importance for the implementation of the hydrogen economy. At present, nearly 95% of the total hydrogen is produced by coal gasification and steam reforming, while only 4% is obtained from water splitting [5]. Currently, hydrogen production through non-renewable resources (i.e., fossil fuels) severely threatens the environment via massive $CO_2$ emission [6]. In addition, steam-methane reforming is an intensive energy consumption process, in which hydrocarbon and water react together at high-temperature to generate $CO_2$ and hydrogen via the following reactions:

$$CH_4 + H_2O \rightarrow CO + 3H_2 \tag{1}$$

$$CO + H_2O \rightarrow CO_2 + H_2 \tag{2}$$

In these reactions, every mole of hydrogen produces one mole of $CO_2$. Neither of these methods for hydrogen production support the reduction of environmental pollution and global warming and, therefore, they are not environmentally benign processes for sustainable hydrogen production. Although electrochemical water splitting is an effective approach to generate cheap and sustainable hydrogen at a large scale, the relatively suppressed efficiency and the lack of efficient, economical, and earth-abundant electrocatalysts for the hydrogen evolution reaction (HER) are major hurdles in this step [7,8]. Electrochemical water splitting was first observed in 1789 yet contributes only 4% of the $H_2$ production worldwide, even after 200 years of development, due to sluggish kinetics of HER at the cathode in water splitting [9]. Nevertheless, similar to ORR, the HER also has to surmount a certain energy barrier (known as overpotential) to occur. Generally, for any electrochemical reaction, the difference between the applied and thermodynamic potentials is termed as overpotential. In this event, for lowering the overpotential and promoting the reaction rate, both ORR and HER usually require the assistance of electrocatalysts. To date, noble metals, particularly Pt-based electrocatalysts, are highly utilized in the practice of boosting the ORR and HER kinetics [10]. However, Pt is restricted by its high cost and scarcity for industry applications [11]. Therefore, the development of highly efficient electrocatalysts with reduced Pt-loading and high utilization is an urgent need for the establishment of a carbon-neutral economy.

Tremendous efforts have been directed in the past decades to potentially address the aforementioned issues. Recent pioneer studies have demonstrated the effective approach to reduce noble metal usage via scaling down the catalyst size to clusters or single atoms [12], incorporating the noble metals with cheap transition metals in the form of alloys [13], and epitaxial or core-shell nanostructures [14,15]. In addition, several studies have highlighted noble metal-free catalysts for ORR and HER including transition metal chalcogenides [16], nitrides [17], phosphides [18], carbides [19], and borides [20]. Moreover, the development of effective metal-free ORR and HER catalysts are also underway [21]. Despite great endeavors, noble metal-free catalysts are still in their infant stage and are not competitive to Pt-based catalysts. Keeping all this in view, herein, we present a mini-review of ORR and HER electrocatalysts, mainly emphasizing noble metal-based heterogeneous nanocatalysts (NCs).

## 2. Reaction Pathways

For designing omnipotent ORR and HER catalysts, it is important to understand how these electrochemical reactions proceed on a catalyst surface. The mechanistic details of ORR and HER are given in the following section.

### 2.1. Mechanism of ORR

The ORR is a complex multi-electron reaction that may include several elementary steps involving different reaction intermediates. The mechanism of ORR is not fully elucidated due to the complexity and changing behavior of the process depending upon the electrode materials, catalysts, and the

electrolyte. However, the literature says that the ORR can proceed either via a direct four-electron transfer route or via a series of two-electron transfer routes irrespective of the electrolyte; the two-electron pathway leads to hydrogen peroxide ($H_2O_2$) production and the four-electron transfer route results in water ($H_2O$) production. The four-electron transfer process is generally preferred because $H_2O_2$ damages the Nafion membrane in fuel cells. The following equations illustrate both of the pathways when using acidic or alkaline electrolytes:

Acidic electrolyte:

$$O_2 + 4H^+ + 4e^- \rightarrow 2H_2O \tag{3}$$

$$O_2 + 2H^+ + 2e^- \rightarrow H_2O_2 \tag{4}$$

$$H_2O_2 + 2H^+ + 2e^- \rightarrow 2H_2O \tag{5}$$

Alkaline electrolyte:

$$O_2 + H_2O + 4e^- \rightarrow 4OH^- \tag{6}$$

$$O_2 + H_2O + 2e+ \rightarrow HO^-_2 + OH^- \tag{7}$$

$$HO^-_2 + H_2O + 2e^- \rightarrow 3OH^- \tag{8}$$

### 2.2. Mechanism of HER

The understanding of the HER mechanism is a key factor for designing the next generation HER catalysts. Depending on the pH values of the electrolyte solution, HER proceeds through the reduction of protons ($H^+$) or water ($H_2O$) molecules accompanied by the subsequent evolution of gaseous hydrogen (Figure 1).

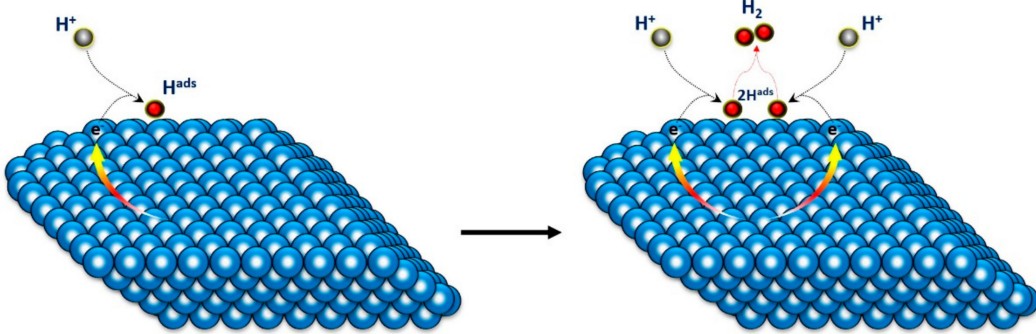

**Volmer-Tafel Mechanism**

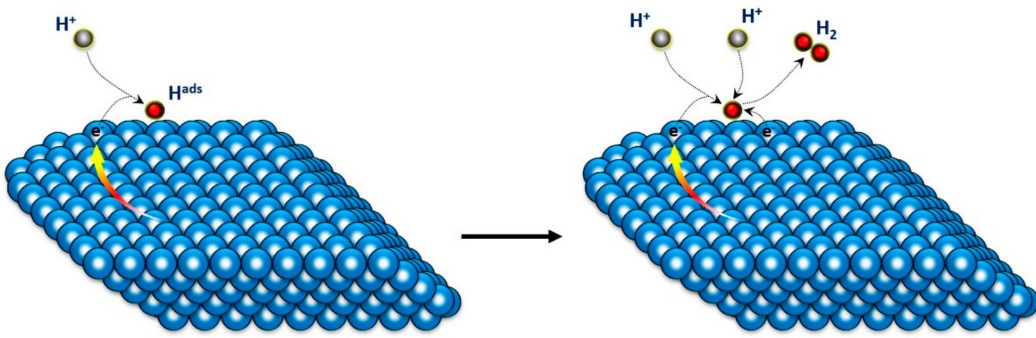

**Volmer-Heyrovsky Mechanism**

**Figure 1.** The hydrogen evolution reaction (HER) pathways on the catalyst surface.

In acidic media (i.e., pH < 7.0), the HER involves two successive steps on the surface of the catalyst. The discharge or Volmer reaction takes place in the first step of HER. In this step, a proton (H+) is adsorbed on the empty active site of the NC surface to yield an adsorbed hydrogen atom ($H^{ads}$) (Equation (9)). The hydronium cation (i.e., $H_3O^+$) is the proton source in the acidic medium. In the subsequent step, a second proton (i.e., $H^+$) combines with an adsorbed hydrogen atom ($H^{ads}$; from step 1) and an electron ($e^-$) to evolve $H_2$ (Equation (10)). This step is known as the Heyrovsky step or electrochemical desorption step or ion + atom step. Herein, $H_2$ formation may also occur via an alternative pathway, which is known as Tafel or chemical desorption step. In this pathway, $H_2$ is produced via a combination of two $H^{ads}$ on the NC surface (Equation (11)).

$$H_3O^+ + e^- \rightarrow H^{ads} + H_2O \tag{9}$$

$$H^{ads} + H_3O^+ + e^- \rightarrow H_2 + H_2O \tag{10}$$

$$2H^{ads} \rightarrow H_2 \tag{11}$$

The HER is relatively slower in an alkaline medium (i.e., pH > 7.0) due to the lack of protons. In an alkaline medium, the HER starts by dissociating water (i.e., $H_2O$) molecules to generate protons. The dissociation of $H_2O$ is involved in both the Volmer (Equation (12)) and Heyrovsky (Equation (13)) steps during HER in an alkaline medium. On the other hand, the Tafel step is the same in an alkaline HER as that in an acidic HER. The overall HER in an alkaline medium is described in Equation (14).

$$H_2O + e^- \rightarrow H^{ads} + OH^- \tag{12}$$

$$H^{ads} + H_2O + e^- \rightarrow OH^- + H_2 \tag{13}$$

$$2H_2O + 2e^- \rightarrow 2OH^- + H_2 \tag{14}$$

It is evident from the aforementioned discussion that additional energy is required to generate protons in alkaline media, and the HER drives slowly in an alkaline medium. Previous studies reported that HER performance in an alkaline medium is significantly controlled by a fine balance between $\Delta G_H°$ and the energy required to dissociate $H_2O$.

## 3. Approaches for Screening the ORR and HER Performances

### 3.1. Evolution of ORR Performance

Commonly, the ORR performance of any electrocatalyst is evaluated by three major aspects: (i) mass activity, (ii) specific activity, and (iii) onset and half-wave potential. The rotating disk electrode (RDE) method is a widely accepted approach to measure the aforementioned parameters. This section includes a detailed discussion of these approaches:

(i) **Mass activity**: The mass activity (MA) is the most important parameter to evaluate the ORR performance of any catalyst, which is the measure of performance per unit loading of metal. Generally, the MA is calculated by the following equation:

$$mass\ activity\ \left( mA\ mg^{-1} \right) = J_k \times \frac{area}{mass\ of\ catalyst} \tag{15}$$

where $J_k$ is the kinetic current density (mA/cm$^2$) and the area is the geometric area of the working electrode. The mass activity of the catalyst is estimated via the calculation of $J_k$ and normalization to the catalyst loading on the glassy carbon rotating disk electrode.

(ii) **Specific activity**: The specific activity (SA) indicates the number of active sites presents on the surface of the catalyst. The electrochemically active surface area (ECSA) plays an important role in evaluating the SA. The ECSA of the experimental catalysts can be calculated by acquiring the

columbic charge for reduction of the monolayer Pt or Pd oxide after integration and double-layer correction using the following equation:

$$ECSA = \frac{Q_{Pt/Pd}}{Q_{ref} \times m} \tag{16}$$

where $Q_{ref}$ denotes the charge required for reduction of the monolayer oxide from the bright Pt surface (i.e., 0.405 mCcm$^{-2}$), m stands for metal loading, and $Q_{Pd}$ represents the charge required for oxygen desorption, which is calculated by the following equation:

$$Q_{Pd} = \frac{1}{v} \int (I - I_d) dE \tag{17}$$

where $v$ is the scan rate for cyclic voltammetry (CV) analysis and integral parts refer to the area under the Pd oxide reduction peak in the CV curves. The kinetic current density ($J_k$) and the number of electrons transferred in ORR are calculated based on the equations below:

$$\frac{1}{J} = \frac{1}{J_K} + \frac{1}{J_L} = \frac{1}{J_K} + \frac{1}{B\omega^{0.5}} \tag{18}$$

$$B = 0.62nFC_{O_2}D_{O_2}^{\frac{2}{3}}v^{-\frac{1}{6}} \tag{19}$$

where $J$, $J_k$, and $J_L$ are the experimentally measured, mass transport free kinetic, and diffusion-limited current densities, respectively, $\omega$ is the angular velocity of the electrode, $n$ is the transferred electron number, $F$ is the Faraday constant, $C_{o2}$ is the bulk concentration of $O_2$, $D_{o2}$ is the diffusion coefficient, and $v$ is the kinematic viscosity of the electrolyte. The SA can be obtained when $J_K$ is normalized to the Pd ECSA.

(iii) **Onset and half-wave potential:** The onset potential (Eoc) and half-wave potential (E$_{1/2}$) are the techniques for the quick screen of ORR performance. Both of these parameters can be obtained from the linear sweep voltammetry (LSV) curve (Figure 2).

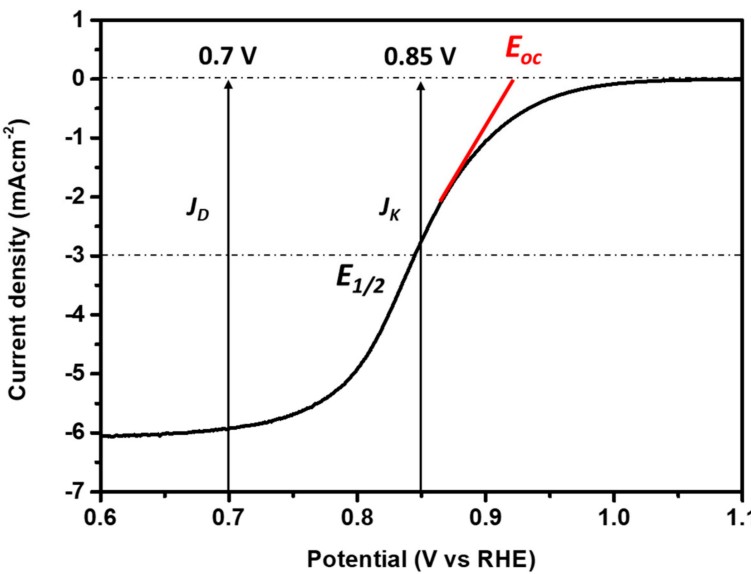

**Figure 2.** The linear sweep voltammetry curve depicting the onset potential (Eoc) and half-wave potential (E$_{1/2}$).

## 3.2. Evolution of HER Performance

To evaluate the HER catalytic performance of NCs, parameters including (i) overpotential, (ii) electrochemically active surface area (ECSA), (iii) Tafel slope, (iv) exchange current density,

(v) Faradic efficiency, (vi) Turnover efficiency, and (vii) stability are taken into consideration. Following are the details of these approaches:

(i) **Overpotential:** The overpotential is the difference between the applied (i.e., experimentally observed potential) and thermodynamically determined potential for any electrochemical reaction. The overpotential is the most important parameter for evaluating the HER performance, and a desired HER catalyst should be capable of lowering the overpotential.

(ii) **Electrochemically active surface area (ECSA)**: ECSA is the active area (i.e., number of active sites) present on the NC surface accessible to the electrolyte for charge transfer. The ECSA can be determined by using electrochemical techniques such as cyclic voltammetry (CV) and CO-stripping analysis.

(iii) **Tafel slope**: Tafel slope, which is actually the slope of the linear region of a Tafel plot (i.e., the plot between overpotential vs. log |current density|) is used to achieve insight into the reaction pathways (mechanisms) of HER on a NC surface. The HER pathways on NC surfaces can be determined by comparing the Tafel slope value with that of theoretical ones. The theoretical Tafel slope values for the Volmer (i.e., the discharge step), the Heyrovsky (i.e., electrochemical desorption step), and the Tafel (i.e., the chemical desorption/combination) steps are 120 mVdec$^{-1}$, 40 mVdec$^{-1}$, and 30 mVdec$^{-1}$, respectively. For instance, previous studies reported that the Tafel slope is 30 mVdec$^{-1}$ for a commercial Pt catalyst in an acidic medium (i.e., 0.5 M $H_2SO_4$), which indicates that HER proceeds through the Volmer–Tafel process on the Pt surface.

(iv) **Exchange current density**: The exchange current density is the current density at the equilibrium potential (i.e., both the cathodic and anodic currents are equal) and reflects the inherent nature of the NC. The exchange current density can be determined via the intersection of the extrapolated linear part of the Tafel plots to the *X*-axis.

(v) **Faradic efficiency**: Faradic efficiency is the ratio of the amount of experimentally produced $H_2$ to the theoretically determined $H_2$ amount.

(vi) **Turnover efficiency**: Turnover efficiency refers to the number of desired product molecules that a NC can produce per catalytic site per unit of time.

(vii) **Stability**: Long term durability/stability is another important aspect for HER catalysts, which describes the ability of a catalyst to maintain its activity when operated for a particular time. The stability can be easily determined by comparing the overpotential and Tafel slopes of the as-prepared condition and after degradation test cycles.

## 4. Noble Metal-Based Electrocatalysts for ORR and HER

As described in the aforementioned section, there are two possible pathways of the HER: (i) Volmer–Tafel and (ii) Volmer–Heyrowsky. Both of these mechanisms involve the adsorption of the hydrogen atom (i.e., $H^{ads}$) on the catalyst surface as an intermediate step, and this ultimately becomes the rate-determining step (i.e., RDS) for the HER kinetics. In this context, the binding energy of $H^{ads}$ on the surface of the catalyst should appropriately balance, meaning that the lower binding energy affects the Volmer step, while strong bond formation severely hampers the desorption process of molecular hydrogen. In brief, the ideal electrocatalysts should have the hydrogen adsorption energies nearly at ($\Delta G_H = 0$) [22]. As widely discussed in the literature, based on this concept, experimental and theoretical studies provide a clear relationship between the HER performance (i.e., exchange current) and the metal-$H^{ads}$ bond strengths (Figure 3a) [23]. Similar to HER, according to the Sabatier principle, the binding energy of ORR intermediate species on the catalyst surface should be appropriately balanced and the relationship between the experimentally measured ORR performance and the calculated $E_{ads}$ of intermediate adsorbates demonstrates a volcano shape, (Figure 3b) [24]. It is evident from Figure 3a,b that Pt stands on top of the volcano plot and thus is considered as the state-of-the-art electrocatalyst to boost ORR and HER kinetics. However, high cost and limited availability are two major obstacles for the commercial viability of Pt-based catalysts. In the past decade, several studies

have highlighted effective strategies to alleviate the economic issues of Pt-based catalysts via reducing the Pt-loading or incorporating Pt with other cheap transition metals in the form of alloys, core-shells, cluster-in-clusters, and/or surface decoration [25]. Meanwhile, significant progress has been made in the Pt-free NCs via incorporating Pd, Au, Ir, and so on with 3d-transition metals in various binary and ternary combinations to improve the kinetics of the reaction. Moreover, the structural and electronic properties of heterogeneous NCs have also been tuned via optimizing the particle size and coupling with different support materials [26–28]. In contrast, despite unprecedented progress in HER and ORR catalysts benefiting from the aforementioned alternative catalyst design strategies, the catalytic performance is still far from practicability and scalability. These approaches are discussed with more details in the following sections.

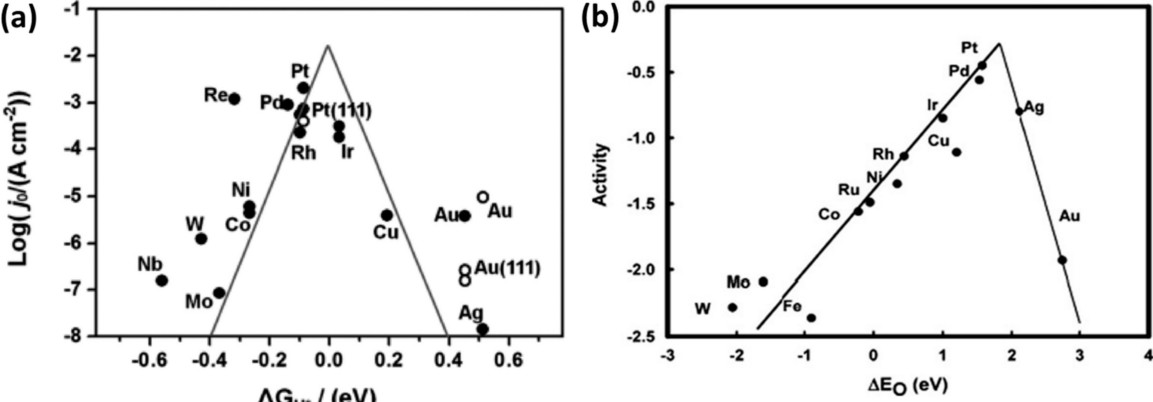

**Figure 3.** (**a**) The relationship between exchange current density ($j_0$) and metal-$H^{ads}$ bond strengths. The volcano shape shows that Pt stands nearly at the middle of the *x*-axis (i.e., $\Delta G_H$) indicating the appropriately balanced $H^{ads}$ bond strength. (**b**) A volcano plot (oxygen reduction reaction (ORR) activity vs. energy of oxygen interaction) indicating the ORR activity of different metals on the (111) surface plane. (Reproduced with permission from [23,24], Royal Society of Chemistry, 2013, American Chemical Society, 2004)

### 4.1. Pt-Based Catalysts in ORR and HER

It is a well-known fact that Pt is by far the most widely accepted catalyst for ORR and HER due to the appropriately balanced adsorption energy of intermediate species on the Pt surface. However, the high materials cost and limited reserves in the Earth's crust are major issues severely hampering its industrial scalability. To this end, there is an urgent need to minimize Pt-loading while maintaining comparable performance. Potential engineering efforts have been directed to surpass the aforementioned deep-rooted issues of Pt-based catalysts in ORR and HER. The electronic structure of Pt strongly affects the adsorption energy of intermediate species during HER and ORR on Pt surfaces. Alloying Pt with foreign transition metals effectively manipulates the electronic structure of surface-active sites, resulting in a downshift in the Pt d-band center, which further weakens the interaction with intermediate adsorbates and largely facilitates the HER and ORR kinetics. Previously reported studies demonstrated effective strategies to manipulate the electronics structure of Pt-based alloys via composition and morphology control [29]. Out of various pioneer studies, $Pt_xCo$ alloys are the most intriguing catalysts for HER and ORR. Oezaslan et al. prepared $Pt_xCo$ alloy nanoparticles in different compositions ($PtCo_3$, $PtCo$, and $Pt_3Co$) on high surface area carbon (HSAC) supports by using a liquid precursor impregnation-freeze-drying-annealing method and demonstrated the structure–performance correlation. The as-prepared $Pt_xCo$ alloys were tested in both alkaline and acidic medium for ORR. Unsurprisingly, the $Pt_xCo$ alloys exhibited different behaviors in alkaline and acidic mediums. The electrochemical active surface areas (ECSA) of $Pt_xCo$ decreased with increasing content of Co in an alkaline medium, while it increased in an acidic medium. The mass activities of $Pt_xCo$ alloys follow different trends in different mediums. In acidic medium, $Pt_xCo$ alloys follow

the trend of PtCo < $Pt_3Co$ < $PtCo_3$, while in an alkaline medium, the trend is $PtCo_3$ < PtCo < $Pt_3Co$, which reveals that ORR performance of $Pt_xCo$ alloys strongly depends upon the composition [30]. The same group also undertook a similar kind of study with $Pt_xCu$ alloys. They prepared $PtCu_3$, PtCu, and $Pt_3Cu$ alloys and found that the mass activities of $Pt_xCu$ alloys followed opposite trends in alkaline and acidic mediums [31].

$Pt_xCo$ nanoparticles are also employed to catalyze HER. Yang et al. synthesized highly efficient and stable PtCo alloy nanoparticles encapsulated in carbon nanofibers (PtCo/CNFs) by integrating the electrospinning and graphitization processes. As-prepared PtCo/CNFs with a Pt-loading of 5 wt.% effectively catalyzed the HER in 0.5 M $H_2SO_4$ solution with a low overpotential and a small Tafel slope of 28 mVdec$^{-1}$ [32]. Alloying with Pt is not limited to Co only. Zhang et al. prepared octahedral Pt–Ni alloy nanoparticles on carbon supports with variable compositions of Pt ($Pt_4Ni$, $Pt_3Ni$, $Pt_2Ni$, $Pt_{1.5}Ni$, and PtNi). Among various compositions of the $Pt_xNi$ alloys, $Pt_{1.5}Ni$ exhibited the highest mass activity of 1.96A/mg$_{Pt}$$^{-1}$ at 0.9 V in ORR [33]. In addition, Du et al. demonstrated that Pt-Ni alloy nanoparticles fabricated on a reduced graphene oxide (rGO) support exhibited excellent electrocatalytic catalytic activity towards HER in alkaline mediums [34]. Apart from controlling the bulk composition, various potential strategies such as thermal annealing have been demonstrated to manipulate the near-surface region of catalysts. For instance, Stamenkovic et al. successfully formed a Pt enriched layer over Pt–M alloy surfaces by using thermal annealing [35]. The Pt-based alloys exhibited enhanced performance towards ORR and HER as compared to that of a commercial benchmark catalyst. The improved activity of Pt-based alloys is due to the change in the electronic structure, i.e., increase of the d-band vacancy, geometric effect, steric effect, and ligand effect [36]. Mainly, whenever an alloy forms, it creates the lattice contraction and eventually leads to changes in the Pt–Pt inter-atomic distance, which favors the dissociative adsorption of oxygen. As a result, it better tunes the ORR activity of alloys compared to pure Pt alone. Unfortunately, despite their good activity, dissolution of the transition metal alloyed in the Pt–M catalysts is a major drawback because these transition metals (M) are electrochemically soluble in low pH conditions. Hence, extensive research is needed to solve the stability issues and their intimate contact within the system.

To surpass the severe issues of Pt-based alloys, such as leaching of non-noble metal and less structural stability in acidic conditions, researchers have come up with a new approach, where, the less stable transition metals are protected with a monolayer (shell) of active Pt. In this way, the problems associated with the dissolution of transition metals are addressed and the interaction of the two metals at the heterogenous binary interface is expected to change the properties of Pt. In such heterogeneous archetypes, a combination of ligand effect (due to electronegativity difference between two atoms) and lattice strain (attributed to the lattice mismatch between intra-particle domains of heterogeneous NCs) tunes the electronic and geometric properties of the material for enhanced ORR activity facilitation [37]. Co-existence of these two effects leads to a stronger electron relocation between the active sites (Pt-shell) and neighboring atoms, which results in a substantially suppressed binding energy of intermediate species, hence, the dramatically enhanced catalytic activity of core-shell structured NCs is obvious. Researchers optimized various compositions and configurations to achieve high performances in the last decades. For example, Wei et al. developed Cu@Pt core-shell electrocatalysts and tested for ORR. As-prepared Cu@Pt NC achieved a peak power density of 0.9 W cm$^{-2}$ at Pt loadings of as low as 0.24 mg cm$^{-2}$ on each cathode and anode [38]. To further decrease the Pt-content, Zhu et al. prepared a Cu@Pt core-shell NC on a Vulcan XC-72R carbon support by using a two-step reduction method, where ethylene glycol (EG) was used as the solvent and reducing agent. As-synthesized NC was employed towards ORR and Cu@Pt NC with an atomic ratio of Cu/Pt of 2.73:1 exhibited tremendous performance [39]. The core-shell structured NCs, in which the active metal such as Pt is decorated only on the surface is also a possibility for increasing Pt-utilization and reducing Pt loading. In this context, Bhalothia et al. also synthesized a similar kind of catalyst with ultra-low loading (5 wt.%) and high utilization of Pt ($Co_{95}Pt_5$) via the deposition-precipitation method in different reduction environments

and found that Co@Pt NC reduced in a CO-environment with only 5 wt.% of Pt surpassed a commercial catalyst and performed in an outstanding manner towards ORR (Figure 4) [40].

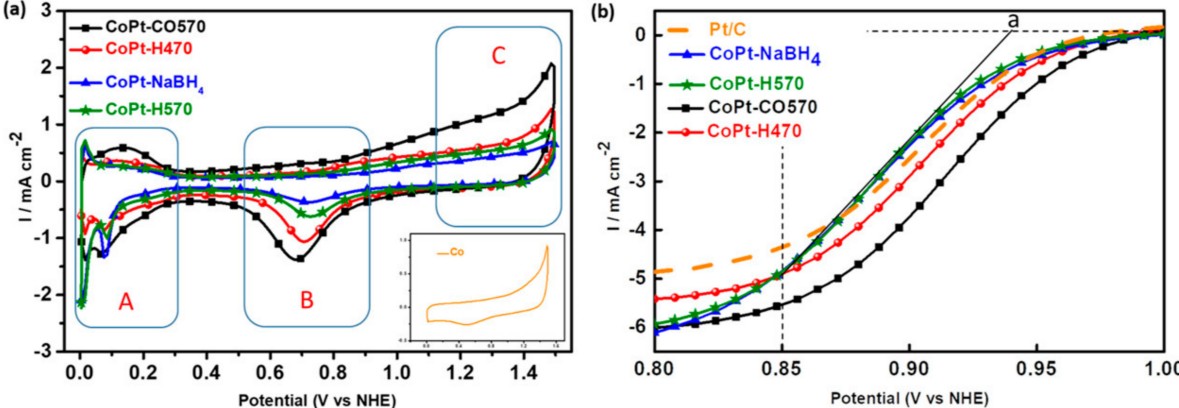

**Figure 4.** (**a**) The cyclic voltammetry and (**b**) linear sweep voltammetry curves of CoPt nanocatalysts (NCs) reduced in different environments. (Reproduced with permission from [40], American Chemical Society, 2019)

In addition to the conventional synthesis methods, Cu@Pt core-shell NCs are also prepared via de-alloying and galvanic replacement processes. To this end, Mani et al. prepared surface de-alloyed Pt-Cu NCs by using the electrochemical method and achieved 4-times improved mass activity as compared to that of the baseline benchmark in ORR [41]. Moreover, Sarkar et al. used the galvanic replacement reaction to prepare carbon-supported Pt@Cu core-shell nanoparticles and as-prepared Pt@Cu NC outperformed the commercial Pt-catalyst [42]. Meanwhile, similar kinds of structures have also been employed to boost the HER kinetics. Hsu et al. used the atomic layer deposition (ALD) technique to deposit Pt on tungsten carbide and produced core-shell structures with Pt-loading reduced by nearly 10-times as compared to that of bulk-Pt catalyst while maintaining similar HER performance [43]. Meanwhile, Qi Shao et al. reported on reduced graphene oxide (rGO) supported AuPt@Pt core-shell nanocrystals to catalyze HER in both acidic and alkaline mediums. As-prepared AuPt@Pt core-shell nanocrystals exhibited improved HER performance with an onset potential ($-25$ mV) and a small Tafel slope (33 mV decade$^{-1}$) in 0.5 M $H_2SO_4$, and an onset potential ($-43$ mV) and small Tafel slope (73 mV decade$^{-1}$) in 0.5 M KOH [44]. Although, core-shell structure NCs seems to be a perfect design, the minimal Pt-loading constraint (i.e., the formation of the Pt-monolayer) and the delicate controllability of perfect core-shell structure with complete Pt-shell coverage is still a challenging task, and can result in structure failure (corrosion of core crystal or segregation of Pt) in the harsh redox conditions. Pt atoms are the only, unified active sites on the core-shell structured NCs, therefore, the selectivity of intermediate species in ORR and HER between surface active sites is suppressed. Consequently, irrespective of the improvement, the unified surface chemical identities affect the traffic of intermediate species and thus suppress the ORR and HER kinetics.

Following this line of research, recent pioneer studies demonstrated that surface decoration of atomic-to-nanoscale clusters on core-shell or cluster-in-cluster structured heterogeneous catalysts further facilitates the kinetics of electrochemical reactions. Taking the advantages of local synergetic collaborations between active sites, Bhalothia et al. decorated highly active atomic Pt-clusters on carbon nanotube-supported Ni@Pd core-shell structure catalysts (NPP) for ORR application. Moreover, they also controlled the depth and distribution of Pt-clusters on the Ni@Pd surface via changing the adsorption time of Pt$^{4+}$ ions. The ORR performance of NPP NCs changed while changing the adsorption time of Pt$^{4+}$ ions due to variation in charge density on the surface. In the optimum case, NPP NCs had the highest specific activity (0.732 mAcm$^{-2}$) when the adsorption time was 2 h. (Figure 5) [45]. The same group has also synthesized tri-metallic NCs consisting of a sub-nanometer Pt-cluster's mask decorated with Pd nano-islands on a NiOx support. They suppressed the Ni-oxide

structure via optimized Pt-loading. In the optimum case, 9 wt.% Pt-decorated Ni@Pd NC achieved the mass activities of 1523.7 mA mg$^{-1}$ and 671.5 mA mg$^{-1}$ at 0.85 V and 0.9 V vs. reversible hydrogen electrode (RHE), respectively [46]. The surface decoration with Pt-atomic clusters not only triggers the ORR activity but also triggers HER activity. Bhalothia et al. also synthesized atomic Pt-cluster decorated Ni@Pd hierarchical structured bimetallic NCs with 2 wt.% Pt-content by using a wet chemical reduction method. As-prepared ternary NC outperformed commercial Pt-catalysts in both alkaline and acidic mediums [47].

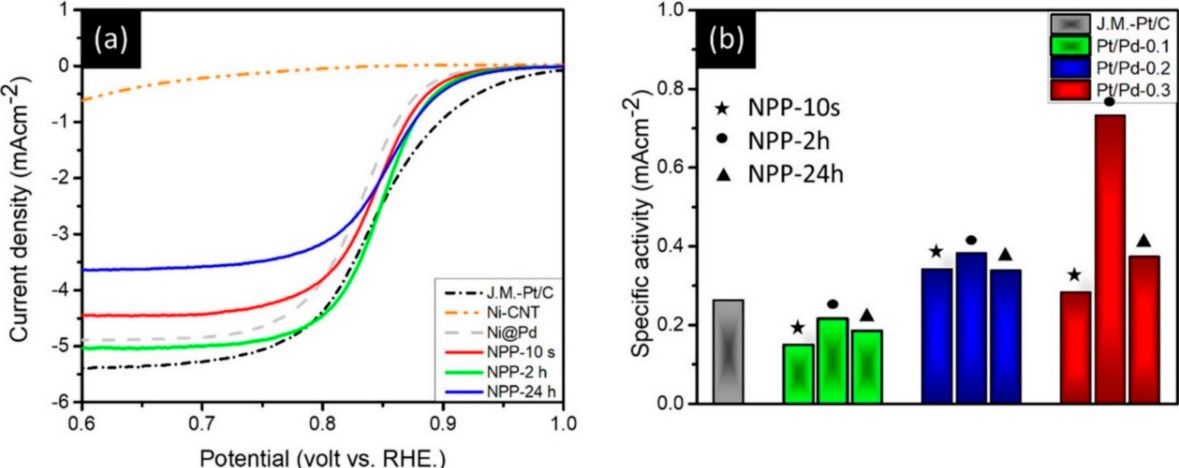

**Figure 5.** (**a**) The linear sweep voltammetry curves and (**b**) corresponding specific activities of nanotube-supported Ni@Pd core-shell structure catalysts (NPP) NCs with different Pt$^{4+}$ adsorption times. (Reproduced with permission from [45], ACS Publications, 2018)

In another study, Bhalothia et al. demonstrated the effect of Pt-metal loading on Cu@Pd core-shell (CuPP) NCs (Figure 6) [48]. The CuPP NCs were synthesized by using a wet chemical reduction method with different Pt loadings of 5 to 14 wt.%. Further, they characterized the CuPP NCs with various X-ray spectroscopic techniques and demonstrated the structural changes with increasing Pt-content. The CuPP NC with 5 wt.% NC produced the optimum mass activity of 639.4 mAmg$^{-1}$, which is 9-times higher as compared to that of a commercial Pt-catalyst. The significantly enhanced performance of CuPP NC with 5 wt.% Pt-content is due to the formation of the truncated surface. Meanwhile, when the Pt-content was raised to 9 wt.%, the surface defects are reduced due to the significant amount of Pt loading. In this event, although the initial activity is suppressed because surface defects are the most active sites for ORR, the stability is improved as compared to that of CuPP NC with 5 wt.% NC.

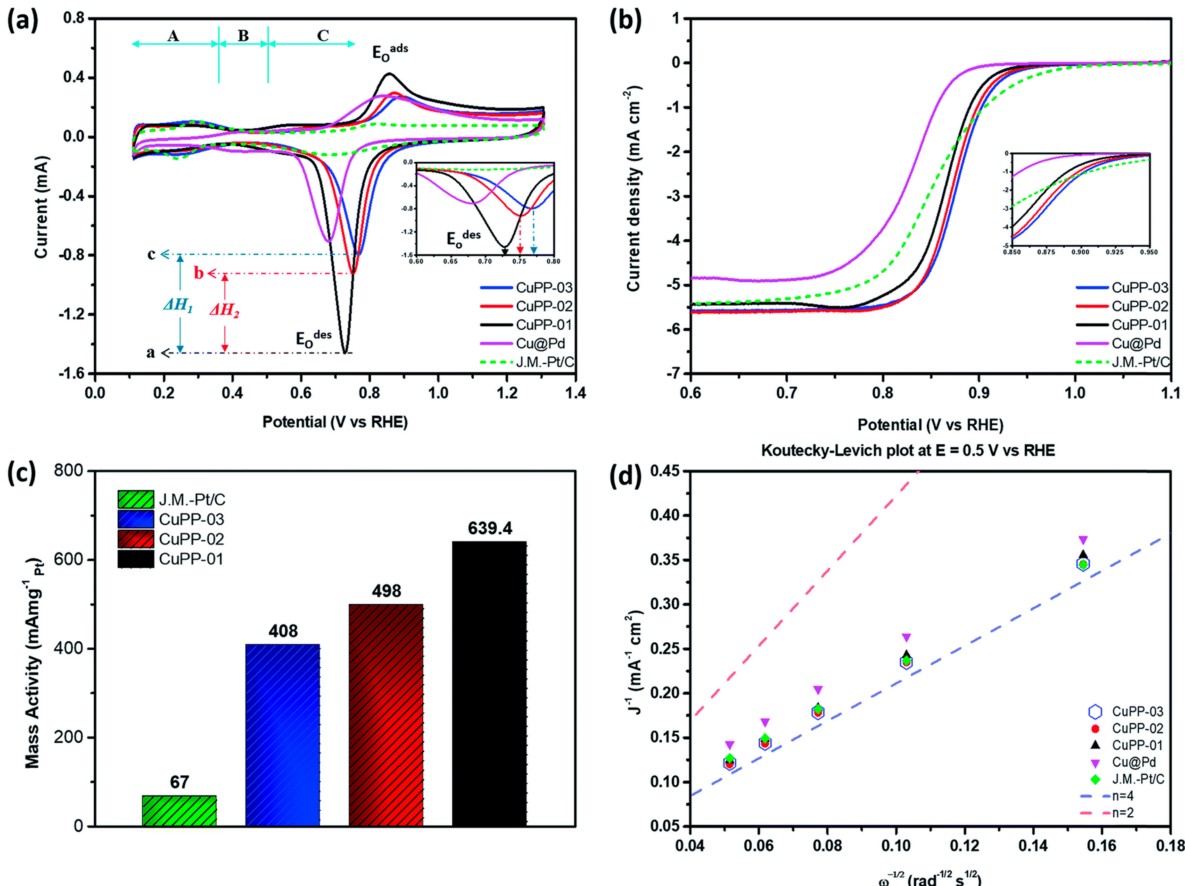

**Figure 6.** (**a**) The cyclic voltammetry, (**b**) linear sweep voltammetry curves, (**c**) ORR mass activities, and (**d**) Koutecky–Levich plot of Cu@Pd core-shell (CuPP) NCs with different Pt-loadings. (Reproduced with permission from [48], Royal Society of Chemistry, 2016)

In a subsequent study, as prepared Pt-cluster decorated Cu@Pd NCs with 14 wt.% of Pt (Cu@Pd/Pt) were subjected to annealing in an $H_2$ environment. After annealing, the specific activity is improved by 2-times as compared to that of the pristine NCs. Of special relevance, the stability of Cu@Pd/Pt is remarkably enhanced after annealing. The current density is increased by 26.3% after 36,000 accelerated degradation test (ADT) cycles (Figure 7) [49]. Bhalothia et al. used Au-clusters to decorate Ni@Pt core-shell catalysts with variable thicknesses of Pt-shells and unveiled the structure–performance relationship of ORR [50]. Of upmost importance, Dai et al. used Pt-trimers to decorate Co@Pd core-shell NCs and demonstrated outstanding ORR performance [51].

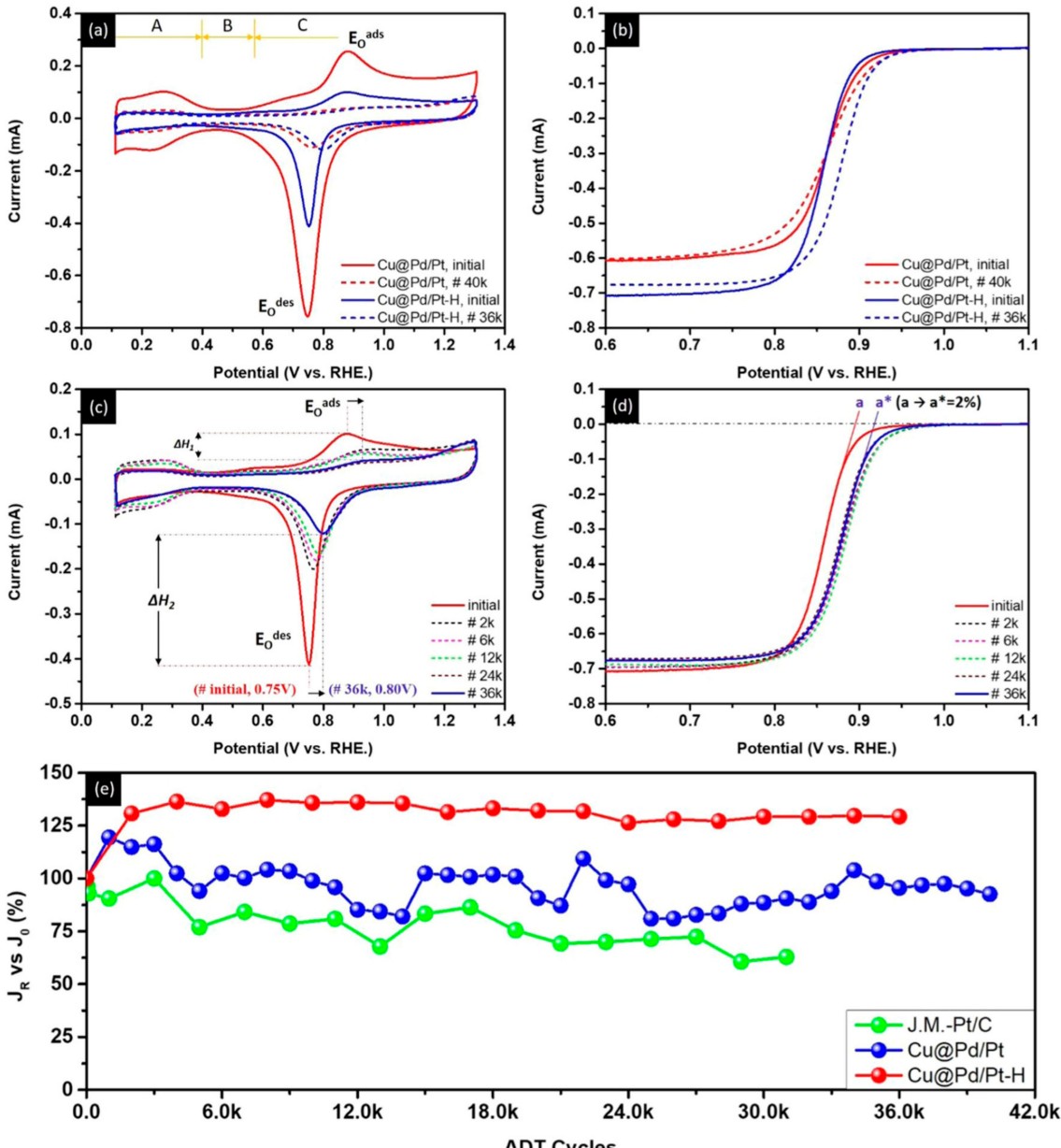

**Figure 7.** (**a**) The cyclic voltammetry and (**b**) linear sweep voltammetry curves of pristine (Cu@Pd/Pt) NCs and NCs after annealing (Cu@Pd/Pt-H). (**c**) The cyclic voltammetry and (**d**) linear sweep voltammetry curves of Cu@Pd/Pt-H NCs at stability test cycles. (**e**) Normalized current densities of Cu@Pd/Pt, Cu@Pd/Pt-H, and commercial J.M.-Pt/C catalysts at selected accelerated degradation test (ADT) cycles. (Reproduced with permission from [49], ACS Publications, 2019)

## 4.2. Pd-Based Catalysts in ORR and HER

Pd-based nanomaterials have emerged as potential contenders to replace Pt-based electrocatalysts. Pd is in the same group as Pt in the periodic table and possesses nearly similar physiochemical properties as that of Pt [52]. In addition, Pd is more economical and abundant than Pt, which might address the cost issues. Although Pd-based nanomaterials are a possible alternative, there is a gap in catalytic performance compared to Pt. Numerous strategies have been adopted to change the electronic structure of Pd for filling this gap. The literature says that catalytic performance of Pd-based catalysts can be significantly improved by several orders via incorporating Pd with late-transition metals in various morphologies and compositions [53], creating atomic vacancies, tuning the size, optimizing support materials, and so on [54,55]. Among the aforementioned approaches, various

studies reported that alloying Pd with foreign transition metals is a widely accepted strategy to tune the electronic properties of Pd for better ORR and HER performance. For example, Bhalothia et al. synthesized Pd nanocrystals adjacent to Ni-oxide in an epitaxial manner with an ordered local structure in both of the Ni and Pd domains [56]. Such a unique nanostructure outperforms a commercial Pt-catalyst with the mass activity of 231.2 mAmg$^{-1}$ and specific activity is 0.492 mAcm$^{-2}$. A similar kind of structure consisting of epitaxially grown Pd nanoparticles on tetrahedral symmetric Ni oxide (NiO$_T$) was synthesized via a wet chemical reduction method on a carbon nanotube (CNT) support. As-prepared NiPd-CNT NC efficiently catalyzed the HER in an acidic medium with an overpotential of 46 mV and a low Tafel slope of 38.0 mV dec$^{-1}$. Of special relevance, the NiPd NC maintained ~95% performance when operated for 1000 cycles (Figure 8) [57].

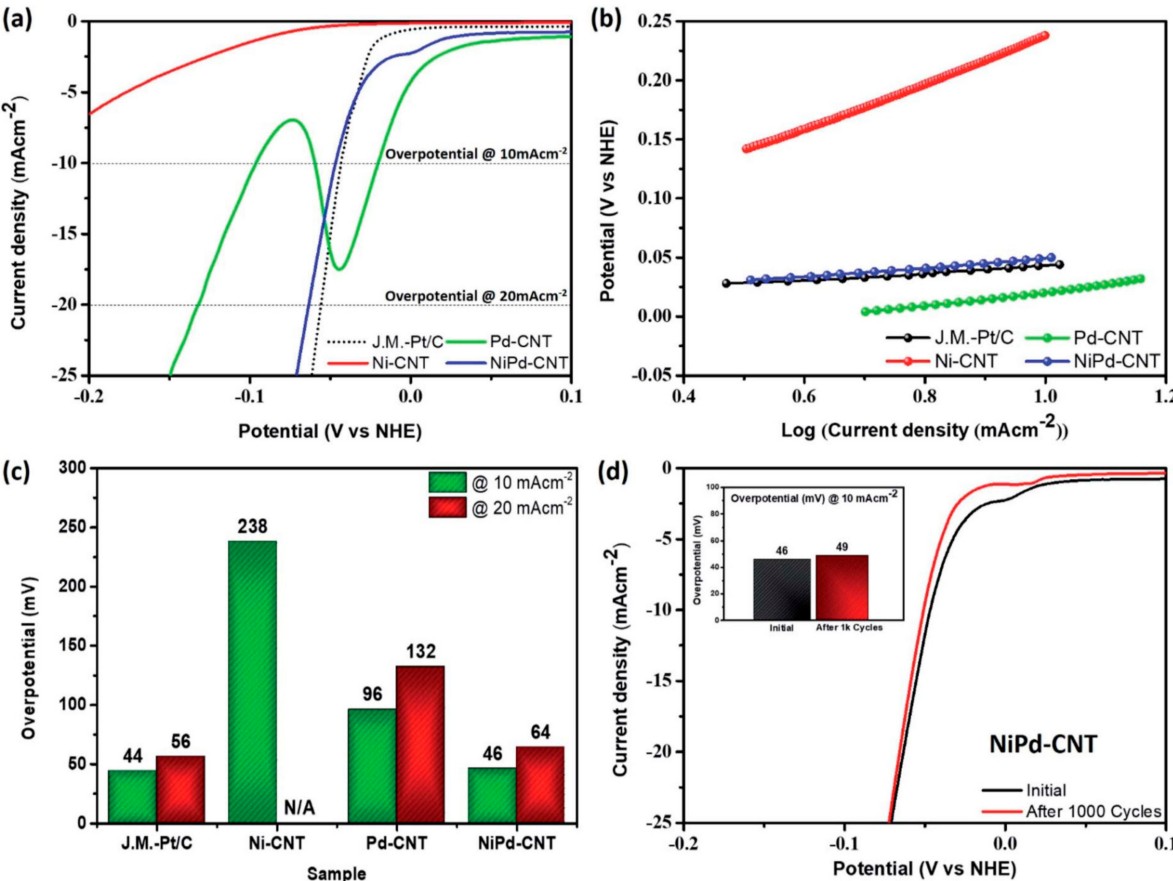

**Figure 8.** (**a**) The linear sweep voltammetry curves, (**b**) Tafel plots, and (**c**) corresponding overpotential values of NiPd-CNT NC compared with those of control samples. (**d**) The stability test results of NiPd-CNT NCs. (Reproduced with permission from [57], Royal Society of Chemistry, 2016)

Zhang et al. synthesized Pd-Co alloy catalysts on carbon supports by using an impregnation method, where sodium borohydride was used as the reducing agent. The ORR performance of as-prepared Pd–Co alloys was further improved by heat treatment in the temperature range of 300 °C to 700 °C. The optimal ORR performance was demonstrated by the Pd-Co alloy catalysts heat-treated at 300 °C [58]. The same group also demonstrated the effect of reducing agents on the morphology and corresponding ORR activity of Pd-Co alloys. Three different reducing agents including ethylene glycol, formaldehyde, and sodium borohydride were utilized to synthesize Pd-Co alloys on the carbon support. Various microscopic and spectroscopic techniques were employed to unveil the morphological differences in Pd–Co alloy nanoparticles synthesized with different reducing agents. These Pd-Co alloys were employed to boost ORR performance and the preferred order was found to be ethylene glycol > sodium borohydride > formaldehyde [59]. The Co-Pd alloy catalysts also

exhibited tremendous activity towards HER. Mech et al. synthesized Co-Pd alloys via co-reduction of $[Co(NH_3)_6]^{3+}$ and $[Pd(NH_3)_4]^{2+}$ complexes for HER. As synthesized Co-Pd alloys achieved a low Tafel slope of 25.4 mVdec$^{-1}$ [60]. Furthermore, Cavallotti et al. prepared the Co–Pd alloys by using the electrochemical deposition of Co/Pd and investigated the HER activity of 0.1 M NaOH electrolytes. As-prepared Co–Pd alloys demonstrated outstanding HER activity [61]. Various studies also reported the effect of support materials on ORR and HER performances of Pd nanoparticles. Ghasemi et al. deposited Pd nanoparticles on graphene via the dip-coating method and investigated their activity towards HER in an acidic medium. The Pd graphene nanocomposite achieved improved catalytic activity as compared to that of Pd and graphene [62]. Andonoglou et al. reported that Pd deposited on activated carbon fibers exhibits outstanding HER performance in an acidic medium [63]. In addition, Huang et al. prepared carbon paper-coated Pd nanoparticles via an electrochemical deposition method. These catalysts with a Pd loading of 0.0106 mg cm$^{-2}$ achieved improved HER performance compared to commercial Pt black electrodes [64]. Zhang et al. demonstrated the particle size effect of Pd nanoparticles towards HER. They prepared Pd nanoparticles on a carbon support within the size range of 3 to 42 nm and tested them in both acidic and alkaline mediums. They observed the similar size effect in both acidic and alkaline mediums, where the exchange current density initially increased from 3 to 19 nm and then reached a similar level as that of bulk Pd [65].

### 4.3. Ir-Based Catalysts in ORR and HER

Iridium (Ir) is also used as an alternative to Pt and Pd-based catalysts owing to its position near the top in the HER volcano plots of both ORR and HER [23,24]. The Ir-based NCs are synthesized in various morphological forms including alloys, core-shells, and even single atoms. Moreover, it is also reported that similar to Pt, Ir is also used for surface decoration of core-shell structured heterogeneous catalysts to boost electrochemical reactions. Sasaki et al. used Ir with Fe as a core-material and prepared IrFe-core@Pt-shell NCs for ORR. The Pt monolayer on IrFe core-shell nanoparticles achieved specific and mass activities of 0.46 mA/cm$^2$ and 1.1 A/mg$_{Pt}$, respectively, in ORR, which are far improved to that of a commercial Pt/C catalyst [66]. Further, Strickler et al. reported a core-shell catalyst where Ir is used as the core material and is covered with platinum (Ir@Pt). The Ir@Pt core-shell catalysts were prepared by a one-pot polyol reduction method with variable thicknesses of Pt-layers. As-prepared Ir@Pt NC exhibits 2.6 and 1.8-times improved specific and mass activities, respectively, compared to a commercial Pt/C (TKK) catalyst [67]. In addition to its use as a core-shell structured catalyst, Ir is also used as surface decoration. Wang et al. synthesized Ir-decorated Pt$_{shell}$–Pd$_{core}$ catalysts on a carbon support and investigated their physical properties and ORR performance. They observed that Pt$_{shell}$–Pd$_{core}$ catalyst safter Ir decoration had better activity as compared to those without decoration. Moreover, the ORR performance of Ir-decorated Pt$_{shell}$–Pd$_{core}$ catalysts also outperformed the commercial catalysts. Such an enhanced ORR performance after Ir-decoration is attributed to the weakened binding energy of Pt–OH during the ORR. The Ir-decorated Pt$_{shell}$–Pd$_{core}$ catalysts achieved a maximum power density of 792.2 mW cm$^{-2}$ at 70 °C in a polymer electrolyte membrane fuel cell (PEMFC), which is about a 24% improvement over that of a commercial Pt/C [68]. Meanwhile, Bhalothia et al. demonstrated that decoration of atomic Ir-clusters significantly enhanced the durability of bimetallic Ni-Pd NCs in ORR [69]. They synthesized tri-metallic NCs consisting of an unconformable Pd-shell over amorphous NiOx with Ir-clusters decoration (NPI) by a self-aligned wet chemical reduction method. The Ir-loading was varied from 1 wt.% (NPI-0025) to 14 wt.% (NPI-14). As-prepared NPI NCs with 1 wt.% of Ir-decoration demonstrated remarkable durability when operated for up to 21 k cycles and retained its 100% performance (Figure 9).

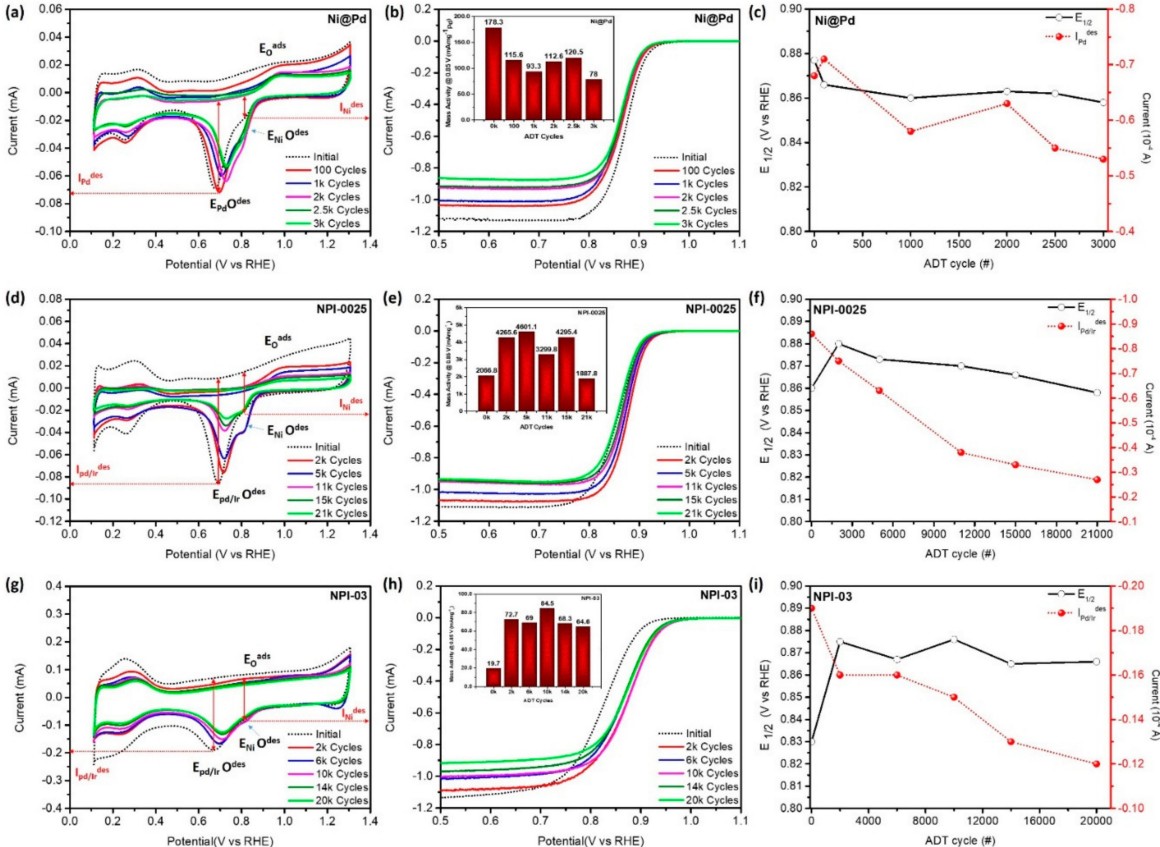

**Figure 9.** The ADT results of NiOx with Ir-clusters decoration (NPI) NCs compared with Ni@Pd. (**a**) cyclic voltammetry and (**b**) linear sweep voltammetry curves of Ni@Pd, (**d**) cyclic voltammetry and (**e**) linear sweep voltammetry curves of NPI-0025 NCs, (**g**) cyclic voltammetry and (**h**) linear sweep voltammetry curves of NPI-03 NCs at selected ADT cycles. The mass activities of corresponding NCs are shown in insets of the linear sweep voltammetry curves, while changes in half-wave potential ($E_{1/2}$) and oxygen desorption current ($I_{Pd}/Ir_{des}$) with increasing ADT cycles are plotted in (**c**), (**f**), and (**i**) for Ni@Pd, NPI-0025, and NPI-03 NCs, respectively. (Reproduced with permission from [69], Elsevier, 2020)

Recently, Ir-single atoms catalysts were shown to be active materials for catalyzing the ORR. Xiao et al. synthesized Ir single-atom catalysts using zeolite imidazolate frameworks-8 (ZIF-8, cavity diameter of 11.6 Å) and found that the as-prepared catalysts exhibited an impressive mass activity of 12.2 A mg$^{-1}$$_{Ir}$ [70].

## 5. Conclusions

An impressive advancement in ORR and HER catalysts has been demonstrated in recent decades. Numerous studies provide the mechanistic understanding of ORR and HER pathways on noble metal-based catalysts, while many of them also exhibited notable control on the desired morphologies, particle size, and compositions. These studies definitely provide a strong foundation for the further development of noble metal-based catalysts with ultra-low loading and high utilization. However, the stability of noble metal catalysts in harsh redox conditions together with the leaching of incorporated transition metals are still challenging tasks in front of the scientific community. Therefore, it is an urgent need to address these issues to compete with conventional energy supply technologies. Development of noble metal-free catalysts is of paramount importance for the wide-spread market introduction of fuel cell and hydrogen technologies.

**Author Contributions:** Formal analysis, D.B. and L.K.; funding acquisition, T.-Y.C.; project administration, T.-Y.C.; resources, T.-Y.C.; supervision, T.-Y.C. and K.-W.W.; validation, C.Y., S.-S.Y., W.-H.H.; writing—original draft, D.B.; writing—review and editing, D.B. and K.-W.W. All authors have read and agreed to the published version of the manuscript.

**Funding:** The APC was funded by the Ministry of Science and Technology, Taiwan (MOST106-2112-M-007-001-MY3, MOST 109-2112-M-007-030-MY3, and MOST109-3116-F-007-001-).

**Conflicts of Interest:** The authors declare no conflict of interest.

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
