# Peer review of "Recent Advancements and Future Prospects of Noble Metal-Based Heterogeneous Nanocatalysts for Oxygen Reduction and Hydrogen Evolution Reactions"

_applsci, doi:10.3390/app10217708_

Round 1

Reviewer 1 Report

The subject of the paper clearly falls within the scope of this Journal.

The paper is very interesting, well written and well organized, and represents some advancement over the actual state-of-the-art. The ways and means are well described as well as the obtained results which are thoroughly discussed and conclusions are well drawn. The paper is also supported by relevant literature review.

I do recommend the publication of this paper.

Author Response

Comment 1:    The paper is very interesting, well written and well organized, and represents some advancement over the actual state-of-the-art. The ways and means are well described as well as the obtained results which are thoroughly discussed and conclusions are well drawn. The paper is also supported by relevant literature review. I do recommend the publication of this paper.

Author reply: The authors appreciate the kind suggestion from the referee. As suggested, the typing mistakes have been corrected in revised manuscript.

Reviewer 2 Report

This manuscript addresses the electrocatalysis of oxygen reduction reaction and hydrogen evolution reaction by noble metal-based heterogeneous catalysts with appropriate literature references. The paper has the appropriate quality to be published in the Applied Sciences journal.

Author Response

Comment 1:   This manuscript addresses the electrocatalysis of oxygen reduction reaction and hydrogen evolution reaction by noble metal-based heterogeneous catalysts with appropriate literature references. The paper has the appropriate quality to be published in the Applied Sciences journal.

Author reply: The authors appreciate the kind suggestion from the referee. As suggested, the typing mistakes have been corrected in revised manuscript.

Reviewer 3 Report

1. It is very useful the section 3 to understand better the development of the materials descripted. The majority of the references used correspond to the last 5 years.  

However, I miss a deeper discussion about  the noble-metal free catalysts. I consider that the authors should add a section about the new tendencies.

2. The manuscript is full of spelling mistakes. It should be carrefully reviewed. 

3. Incorrect use of subscripts and superscripts in:

- Equation 5

- Line 122

- Line 131

4. There is an incorrect use of capital letters and lowercase letters in: 

- Line 141 performance instead of Performance

- Line 201 HER instead of Her

- Line 236 enormous instead of Enormous

- Line 405 Pd instead of pd

- Line 490 the instead of The

5. Relevant conceptual problems with adsorption and absorption: 

Lines 222, 224, 229, 247 and 254 Adsorption instead of absorption

Author Response

Comment 1:   It is very useful the section 3 to understand better the development of the materials descripted. The majority of the references used correspond to the last 5 years. However, I miss a deeper discussion about the noble-metal free catalysts. I consider that the authors should add a section about the new tendencies.

Author reply: The authors appreciate the kind suggestion from the referee. In the present review article, we specifically focused on the recent advancements and future prospects of noble metal-based heterogeneous nanocatalysts for oxygen reduction reaction (ORR) and hydrogen evolution reaction (HER). Moreover, previously published review articles based on the noble-metal free catalysts particularly focused on the individual groups such as transition metal nitrides (Reference: Current Opinion in Solid State & Materials Science 24 (2020) 100805), transition metals phosphides (Reference: Adv. Energy Mater. 2015, 5, 1500985) etc. Therefore, we regret that combining the discussion about noble-metal free catalysts is not possible in current review. We can write another review article based on noble-metal free nanocatalysts

Comment 2:   The manuscript is full of spelling mistakes. It should be carefully reviewed. 

Author reply: The authors appreciate the kind suggestion from the referee. As suggested, the typing mistakes have been corrected in revised manuscript.

Comment 3:   Incorrect use of subscripts and superscripts in:

- Equation 5

- Line 122

- Line 131

Author reply: The authors appreciate the kind suggestion from the referee. As suggested, the typing mistakes have been corrected in revised manuscript.

Comment 4:   There is an incorrect use of capital letters and lowercase letters in: 

- Line 141 performance instead of Performance

- Line 201 HER instead of Her

- Line 236 enormous instead of Enormous

- Line 405 Pd instead of pd

- Line 490 the instead of The

Author reply: The authors appreciate the kind suggestion from the referee. As suggested, the typing mistakes have been corrected in revised manuscript.

Comment 5:   Relevant conceptual problems with adsorption and absorption: 

Lines 222, 224, 229, 247 and 254 Adsorption instead of absorption

Author reply: The authors appreciate the kind suggestion from the referee. As suggested, the typing mistakes have been corrected in revised manuscript.